# Patterns of respiratory health services utilization from birth to 5 years of children who experienced adverse birth outcomes

Jesus Serrano-Lomelin[1], Anne Hicks[2], Manoj Kumar[2], David W. Johnson[3], Radha Chari[1], Alvaro Osornio-Vargas[2], Susan Crawford[4], Jeffrey Bakal[4], Maria B. Ospina[1]*

1 Faculty of Medicine & Dentistry, Department of Obstetrics & Gynecology, University of Alberta, Edmonton, Alberta, Canada, 2 Faculty of Medicine & Dentistry, Department of Pediatrics, University of Alberta, Edmonton, Alberta, Canada, 3 Department of Pediatrics, Emergency Medicine, and Physiology & Pharmacology, Cumming School of Medicine, University of Calgary, Calgary, Alberta, Canada, 4 Alberta Health Services, Edmonton, Alberta, Canada

* mospina@ualberta.ca

## Abstract

### Introduction

Adverse birth outcomes have important consequences for future lung health. We evaluated patterns of respiratory health services utilization in early childhood among children born preterm (PTB), small and large for gestational age at term (SGA and LGA, respectively), and appropriate-for-gestational age at term.

### Materials and methods

We conducted a population-based retrospective cohort study using administrative health data of all singleton live births in Alberta, Canada between 2005–2010. Data on hospitalizations and emergency department (ED) visits from birth to 5 years were collected for asthma, bronchitis, bronchiolitis, croup, influenza, pneumonia, and other acute upper and lower respiratory tract infections (other URTI and other LRTI, respectively). Adjusted rate ratios were estimated for respiratory ED visits and hospitalizations for adverse birth outcomes using the appropriate-for-gestational age at term group as reference. Age-specific trajectories of total respiratory health services utilization rates for each group were estimated in Poisson models.

### Results

A total of 293,764 episodes of respiratory care from 206,994 children were analyzed. Very PTB children had the highest rates of health services use for all respiratory conditions, particularly for asthma, pneumonia, and bronchiolitis hospitalizations. Moderate/late PTB children also had elevated ED visits and hospitalizations for all respiratory conditions. Children born SGA showed high rates of ED visits for other LRTI, and of hospitalizations for bronchitis, bronchiolitis, and other URTI. Children born LGA had high rates of croup and other URTI ED visits, and of bronchiolitis and bronchiolitis hospitalizations. Age-specific trajectories

**Data Availability Statement:** Data cannot be shared publicly because it is held securely in coded form at Alberta Health Services. Alberta Health

Services is the legal custodian of the original data. Alberta Health Services' policies and acts (e.g., Health Information Act of Alberta) guarantee the security, privacy and confidentiality of the patient data. Data agreement with Alberta Health Services prohibits researchers from making the dataset publicly available. Access to data may be granted to those who meet pre-specified criteria for confidential access. Data are available from Alberta Health Services Provincial Research Data Services for researchers who meet the criteria for access to confidential data. The data underlying the results presented in the study are available from Alberta Health Services' (AHS) Health System Access (HSA): https://www.albertahealthservices.ca/research/page8579.aspx. More information at: research.administration@ahs.ca.

**Funding:** This research was funded by the Lois Hole Hospital for Women through a Women and Children's Health Research Institute Recruitment Award (MBO) (https://www.wchri.org), and the Lung Association Alberta & NWT through the 2017-2018 National Grant Review program (https://www.ab.lung.ca/what-we-do/research/grant-opportunities/national-grant-review). MBO's research is supported by the Canada Research Chair Program (Government of Canada; Ottawa, Canada). The funders had no role in study design, data collection and analysis, decision to publish, or preparation of the manuscript.

**Competing interests:** The authors have declared that no competing interests exist.

showed a decreasing trend in the rates of total respiratory health service utilization from birth to five years of age for all groups studied. Children born PTB and LGA at term significantly required more respiratory health services over time compared to the reference group.

## Conclusion

Patterns of paediatric respiratory health services utilization vary according to gestational age and fetal growth.

## Introduction

Alterations in fetal growth and duration of gestation are adverse birth outcomes that increase the risk of respiratory diseases both in childhood and adult life [1–6]. Much of the evidence to date about the relationship between adverse birth outcomes and lung problems in childhood has focused on the associations between preterm birth (PTB) or low birth weight and a high risk of asthma and asthma-like symptoms [5–9]. The increased susceptibility to respiratory diseases among children born PTB has been linked to the immaturity of both respiratory and adaptive immune systems at birth [10].

There is limited evidence about how other adverse birth outcomes (i.e., small and large for gestational age [SGA, LGA]) impact respiratory diseases other than asthma or asthma-like symptoms in early childhood. Physiological mechanisms linking SGA and LGA to future respiratory diseases are unclear and conflicting evidence has been reported for the association between SGA and LGA and respiratory health in childhood. For SGA, insufficient input of oxygen and metabolites linked to fetal growth restrictions could negatively impact the lung development of the fetus [2]. For LGA, both the higher risk of experiencing respiratory distress syndrome [11] and the reduced lung functional capacity observed in obese infants may alter the risk of suffering respiratory problems [12].

Some studies have reported that children born SGA have an elevated risk of asthma at ages 3 to 18 years [13], and an increased number of respiratory hospitalizations before the age of five [14]. Other studies have reported inconclusive evidence of associations between being born SGA at term or preterm and asthma or bronchitis/pneumonia symptoms at 5 years of age [15], or with respiratory viral infections during the first 6 months of life [16].

Paediatric respiratory diseases are major causes of morbidity and mortality worldwide [17]. In Canada, approximately 15% of children aged 4 to 11 years are diagnosed with asthma every year [18] while croup affects about 6% of children under six years of age [19]. Along with the high prevalence of pediatric respiratory diseases in Canada, the prevalence of PTB is 8% and fluctuates around 10% for SGA and LGA [20]. The health care costs during the first ten years of life for children who experienced PTB is high in Canada [21]. Knowledge gaps remain about the burden imposed by SGA and LGA on health care systems in relation to paediatric respiratory morbidity, especially for diseases other than asthma. Previous studies evaluating health care services utilization for respiratory diseases in children experiencing adverse birth outcomes have primarily focused on hospitalizations [13–15,22–24], whereas patterns of emergency department visits (ED visits) have seldom been explored [25].

To our knowledge, there are no previous population-based cohort studies examining and comparing patterns of respiratory health services utilization (ED visits and hospitalizations) among children who experienced alterations in fetal growth and duration of gestation. The objectives of this study were: (1) to evaluate patterns of ED visits and hospitalizations up to 5

years of age for a broad range of respiratory diseases among children who experienced adverse birth outcomes; and (2) to compare age-specific trajectories of total respiratory health services utilization across adverse birth outcomes. Results from this study may improve our understanding of the relative importance of alterations in fetal growth and duration of gestation in early childhood healthcare pathways.

## Materials and methods

### Study design

This population-based retrospective birth cohort study used provincial health data from Alberta, a culturally diverse province located in Western Canada with a population of ~ 4 million people [26] and a universal single-payer health care system. The study is reported following the STrengthening the Reporting of OBservational studies in Epidemiology (STROBE) guidelines [27].

### Study population

The birth cohort consisted of all singleton live births from deliveries ($\geq$22 weeks of gestation) that occurred between April 1, 2005 and March 31, 2010 in hospitals or attended by registered midwives at home in Alberta. Stillbirths, births that occurred outside of Alberta and multiple births were excluded. Multiple births, representing a small proportion of the total births (approximately 2.6%), were excluded as their neonatal and childhood outcomes are known to differ significantly as compared to the larger proportion of the singleton live births.

### Data sources

The birth cohort was identified from the Alberta Perinatal Health Program (APHP), a validated clinical perinatal registry that collects information on maternal demographic and delivery characteristics, pregnancy outcomes, and newborn's health status for all births occurring in hospitals or attended by registered midwives at home in Alberta [28]. Deaths and stillbirths were identified from Alberta Vital Statistics. Data on health services utilization from birth to five years of age were obtained from the National Ambulatory Care Reporting System (NACRS) and the Discharge Abstracts Database (DAD) for ED visits and hospitalizations, respectively. Both NACRS and DAD record the episodes of care using the International Classification of Diseases, 10th Revision, enhanced Canadian version (ICD-10-CA) [29]. The data were extracted by Alberta Health Services using unique identifiers to link data across health administrative datasets. All final patient records were de-identified before data files were accessed by the authors for data analysis.

### Exposures and outcome measures

**Preterm birth, SGA at term and LGA at term.**   Birth cohort members were classified into three exposure groups: (1) PTB (both spontaneous and induced), defined as a live birth with a gestation period < 37 weeks and sub-classified as moderate/late PTB (32–36 complete weeks of gestation) and very PTB (< 32 weeks of gestation) [30]; (2) SGA at term, defined as a live birth with $\geq$ 37 weeks of gestational age and a weight below the 10th percentile for gestational age and sex as per the 2001 Canadian population growth charts [31]; and (3) LGA at term, defined as a live birth with $\geq$ 37 weeks of gestational age and weight above the 90th percentile for gestational age and sex as per the 2001 Canadian population growth charts [31]. We focused on SGA and LGA at term to differentiate from PTB, a condition in which lung development immaturity has been well documented [10]. Members of the birth cohort that did not

experience any of the study exposures were the reference group (i.e., appropriate-for-gestational age infants born at term).

## Study outcomes

**Respiratory health services utilization.** Respiratory health services utilization was defined as all ED visits and hospitalizations that occurred from birth to 5 years of age with an ICD-10-CA primary diagnostic code indicative of acute bronchitis (J20), bronchiolitis (J21), asthma (J45), croup (J05), influenza (J09-J11), pneumonia (J12-J18), other acute lower respiratory tract infections (other LRTI) (J22), and other acute upper respiratory tract infections (other URTI) (J00-J06, except J05). Recurrent wheezing (R06.2) events were merged with asthma or bronchiolitis based on the most prevalent condition after the first wheezing episode. This merge was made because early life diagnosis of asthma and/or bronchiolitis is particularly difficult in acute settings, and recurrent wheezing has been associated with the development of asthma and respiratory viral infections [32–34]. Data were censored at date of death or end of the follow-up period (i.e., 5 years of age).

## Other variables

Clinical and sociodemographic characteristics related to pediatric respiratory problems were considered for statistical analysis. They included bronchopulmonary dysplasia (BPD; yes/no), Apgar score at 5 minutes (classified as low [0–3], moderately abnormal [4–6], and reassuring [7–10]) [35], sex recorded at birth (female, male), the use (yes/no) of significant resuscitative measures at birth (bag/mask, cardiopulmonary resuscitation, endotracheal tube or epinephrine), and maternal socioeconomic status (SES) at delivery.

We used the Pampalon Material and Social Deprivation Index as a proxy measure of SES. The Pampalon Index is a nationwide area-level composite indicator that integrates 2006 Canadian census data by dissemination area (the smallest standard geographic area for which census data is disseminated) regarding income, education, employment (for the material component), marital status, one-person household, and single-parent families (for the social component) for the population aged 15 and over [36]. The Pampalon Index has been used in previous Canadian studies as a valid measure of area-level SES [37]. The material and social deprivation components of the Index are reported in quintiles, where $Q_1$ and $Q_5$ correspond to the least and most deprived groups, respectively. The six-character maternal postal codes at delivery were geographically linked to the dissemination areas (which are a conglomerate of postal codes), as reported elsewhere [38].

The previous clinical and sociodemographic characteristics were treated as risk factors associated with respiratory problems during early childhood stages regardless of their role in causal pathways (i.e., confounders, moderators, or effect-modifiers). Our comparative analysis was not aimed to formally test causal pathways, but in generating measures of association balanced for other known risk factors. Moreover, the essential role of socioeconomic in producing respiratory health differences for each respiratory condition included in this study has been previously evaluated [39].

## Statistical analysis

Baseline demographic and clinical characteristics were described using frequencies and percentages. Counts of ED visits and hospitalizations were tabulated. Crude rates of ED visits and hospitalizations were calculated for each respiratory condition by adverse birth outcome. The total number of episodes of respiratory care was used as the numerator, and the total number

of singleton live births in each group as denominator. Rates were expressed as episodes of respiratory care per 1,000 singleton live births up to five years of age.

We used random intercept coefficient Poisson regression models to evaluate patterns of ED visits and hospitalizations up to 5 years of age for each respiratory condition. Crude and adjusted rate ratios (RR and aRR, respectively) with 95% confidence intervals (CI) were calculated in relation to the reference group. Separate models for ED visits and hospitalizations were estimated for each respiratory disease. The dependent variable was the number of respiratory events from birth to five years of age. Independent variables were the adverse birth groups adjusted by relevant clinical and sociodemographic factors (i.e., sex, 5-minute Apgar score, bronchopulmonary dysplasia, use of significant resuscitation methods, and material and social deprivation). The random-intercept coefficients accounted for area-level variations in the DA for which material and social deprivation indexes were reported.

Using a hierarchical longitudinal Poisson model, we estimated age-specific trajectories of total respiratory health services utilization up to five years of age for the combined respiratory conditions by each birth group. ED visits and hospitalizations occurring on the same day were counted as one event in the trajectories analysis. The dependent variable was the total number of respiratory episodes of care (both hospitalizations and ED visits combined) per year of life. Independent variables were the adverse birth group, year of age (1 to 5), and an interaction term for adverse birth group and year of age. The model was adjusted for baseline factors (i.e., sex, bronchopulmonary dysplasia, Apgar 5-score, use of significant resuscitative measures, and material and social deprivation). Random intercepts were also considered at the DA level. After calculating incidence RR from the models, we estimated marginal respiratory care utilization rates with 95% CI per adverse birth group for the ages one to five. We graphically displayed the longitudinal trajectories of health services utilization rates as number of episodes of respiratory health services utilization every year of life per 1,000 singleton live births. Finally, for each year of life, we estimated the difference in marginal rates between the adverse birth group and the reference group using contrast tests with Bonferroni's correction to adjust the 95% CIs. We used the log number of years of follow-up as offset in all Poisson models to correct for unequal lengths of follow-up due to censoring before the age of five. Missing data were not replaced in the analyses. Age-specific trajectories of respiratory health services utilization for single respiratory conditions by birth groups were estimated and reported in the supplementary material (Tables A to H in S1 Appendix). Statistical analyses were conducted using Stata 15.1 [40].

### Ethics statement

We used de-identified data from Alberta Health Services, the sole provider of health services in the province and legal custodian of the data. Alberta Health Services' policies and acts protect the security, privacy, and confidentiality of patient data collected. The anonymity of cases was guaranteed using the dissemination areas as the geographic reference instead of the maternal postal code at delivery. Therefore, informed consent was not necessary. The research project was approved by the University of Alberta's Health Research Ethics Board (Pro00081365).

## Results

### Demographics and clinical characteristics

The birth cohort consisted of 206,994 live singleton births (Fig 1), which represent 96.6% of the total births registered in the province during the study period.

Demographic and clinical characteristics of the study population are shown in Table 1. The prevalence of PTB in the birth cohort was 9.2% (moderate/late PTB 7.7%, and very PTB 1.5%),

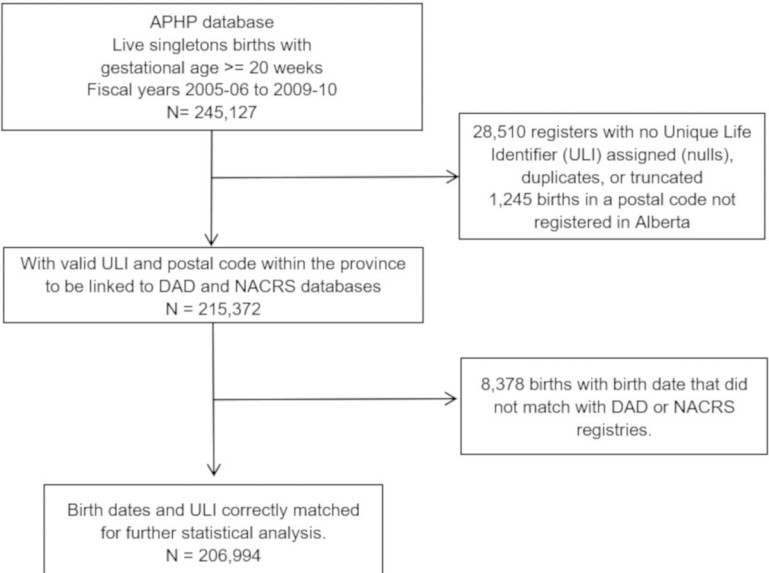

**Fig 1. Flow diagram of assembly of the study cohort.**

while 7.9% and 8.8% were SGA and LGA at term, respectively. There were 183 (0.1%) deaths of birth cohort members at follow-up.

## Rates of respiratory health services utilization

A total of 293,764 episodes of respiratory care (276,293 ED visits and 17,471 hospitalizations) was made by the 206,994 members of the birth cohort during the 5-year follow-up period. Ninety-four percent of the episodes of care were registered as ED visits while 6% were hospitalizations. Fifty-eight percent of children had two or more ED visits, while 33% had two or more hospitalizations during the study period. The number of events per child until they turned five years of age are shown in the supplementary S1 and S2 Figs.

Overall, the highest ED visit rates in all study groups were attributed to other URTI, followed by croup, asthma, and pneumonia (Table 2). Hospitalizations in all study groups were mostly for bronchiolitis and pneumonia. The least frequent reasons for ED visits were influenza and other LRTI, whereas influenza and bronchitis were the least frequent reasons for hospitalizations, compared to all other respiratory diseases.

## Multivariable analysis of respiratory health services utilization

Figs 2 and 3 display adjusted RRs for respiratory hospitalizations and ED visits during the first five years of life for adverse birth groups (all unadjusted and adjusted RRs are presented in S1 Table). Compared to the reference group, children born PTB had the highest rates of respiratory hospitalizations and ED visits, followed by LGA at term and SGA at term.

Compared to the reference group, children born moderate/late PTB had significantly higher ED visits and hospitalization rates for all seven respiratory diseases studied. Increases in ED visit rates ranged from 1.2 times for other URTI (aRR 1.15; 95% CI 1.10, 1.19) to 1.7 times for bronchiolitis (aRR 1.67; CI 1.56, 1.80). Increased hospitalization rates among children born moderate/late PTB ranged from being 1.5 times higher for croup (aRR 1.52; 95% CI 1.20, 1.92) to 2.4 times higher for other LRTI (aRR 2.41; CI 1.77, 3.28) relative to the reference group.

**Table 1. Characteristics of the study population: Singleton live births in Alberta from April 1, 2005 to March 31, 2010.**

| Characteristic | N | % |
|---|---|---|
| Live singletons | **206,994** | **100.0** |
| Sex | | |
|     Female | 100,795 | 48.7 |
|     Male | 105,967 | 51.2 |
|     Missing data | 232 | 0.1 |
| Adverse birth group | | |
|     Reference group | 149,743 | 72.3 |
|     Moderate/late PTB (32–26) | 15,861 | 7.7 |
|     Very PTB ($<$32) | 3,107 | 1.5 |
|     SGA at term | 16,241 | 7.9 |
|     LGA at term | 18,134 | 8.8 |
|     Missing data | 3,908 | 1.9 |
| 5-min Apgar | | |
|     Reassuring (7–10) | 194,943 | 94.2 |
|     Moderately abnormal (4–6) | 3,552 | 1.7 |
|     Low (0–3) | 832 | 0.4 |
|     Missing data | 7,667 | 3.7 |
| Significant resuscitation measures | 18,942 | 9.2 |
| Bronchopulmonary dysplasia | 128 | 0.1 |
| Material deprivation (quintiles) | | |
|     $Q_1$ (least deprived) | 39,806 | 19.2 |
|     $Q_2$ | 38,931 | 18.8 |
|     $Q_3$ | 39,007 | 18.8 |
|     $Q_4$ | 37,635 | 18.2 |
|     $Q_5$ (most deprived) | 43,193 | 20.9 |
|     Missing data | 8,422 | 4.1 |
| Social Deprivation (quintiles) | | |
|     $Q_1$ (least deprived) | 27,353 | 13.2 |
|     $Q_2$ | 39,990 | 19.3 |
|     $Q_3$ | 43,927 | 21.2 |
|     $Q_4$ | 46,120 | 22.3 |
|     $Q_5$ (most deprived) | 41,182 | 19.9 |
|     Missing data | 8,422 | 4.1 |
| Deaths at follow-up | 183 | 0.1 |

Reference group = appropriate-for-gestational age infants born at term; LGA = large for gestational age.
PTB = preterm birth. SGA = small for gestational age.

Children born very PTB had significantly increased ED visit rates for croup (aRR 1.24; 95% CI 1.04, 1.48), bronchitis (aRR 1.35; 95% CI 1.05, 1.73), bronchiolitis (aRR 2.20; 95% CI 1.86, 2.61), pneumonia (aRR 2.26; 95% CI 1.90, 2.69), and asthma (aRR 2.43; 95% CI 2.01, 2.93). Children born very PTB had increased hospitalization rates for all seven respiratory conditions evaluated, ranging from 2.3 times for influenza (aRR 2.33; 95% CI 1.30, 4.18) to near 5 times for other LRTI (aRR 4.81; 95% CI 2.90, 7.97).

Children born SGA at term had significantly higher ED visit rates for other LRTI (aRR 1.16; 95% CI 1.01, 1.32) but lower ED visit rates for croup (aRR 0.86; 95% CI 0.82, 0.92)

**Table 2. Crude rates of ED visits and hospitalizations per 1,000 singleton live birth for respiratory health services from birth to 5 years in Alberta.**

| | Total | | Reference group | | Moderate/late PTB | | Very/extremely PTB | | SGA at term | | LGA at term | |
|---|---|---|---|---|---|---|---|---|---|---|---|---|
| | Events | Rate | Events | Rate | Events | Rate | Events | Rate | Events | Rate | Events | Rate |
| **ED visits** | | | | | | | | | | | | |
| URTI | 150,551 | 741.3 | 109,076 | 728.4 | 12,935 | 815.5 | 2,261 | 727.7 | 11,264 | 693.6 | 15,015 | 828.0 |
| Croup | 33,072 | 162.8 | 23,819 | 159.1 | 3,031 | 191.1 | 588 | 189.3 | 2,195 | 135.2 | 3,439 | 189.6 |
| Asthma | 25,593 | 126.0 | 17,718 | 118.3 | 2,841 | 179.1 | 829 | 266.8 | 2,025 | 124.7 | 2,180 | 120.2 |
| Pneumonia | 22,342 | 110.0 | 15,419 | 103.0 | 2,417 | 152.4 | 831 | 267.5 | 1,651 | 101.7 | 2,024 | 111.6 |
| Bronchiolitis | 19,302 | 95.0 | 13,075 | 87.3 | 2,396 | 151.1 | 618 | 198.9 | 1,414 | 87.1 | 1,799 | 99.2 |
| Bronchitis | 14,015 | 69.0 | 10,098 | 67.4 | 1,264 | 79.7 | 311 | 100.1 | 923 | 56.8 | 1,419 | 78.3 |
| Influenza | 6,858 | 33.8 | 4,956 | 33.1 | 655 | 41.3 | 98 | 31.5 | 517 | 31.8 | 632 | 34.9 |
| LRTI | 4,560 | 22.5 | 3,171 | 21.2 | 475 | 29.9 | 93 | 29.9 | 371 | 22.8 | 450 | 24.8 |
| **Hospitalizations** | | | | | | | | | | | | |
| Bronchiolitis | 5,109 | 25.2 | 3188 | 21.3 | 754 | 47.5 | 303 | 97.5 | 387 | 23.8 | 477 | 26.3 |
| Pneumonia | 4,906 | 24.2 | 3175 | 21.2 | 612 | 38.6 | 336 | 108.1 | 358 | 22.0 | 425 | 23.4 |
| Asthma | 2,984 | 14.7 | 1962 | 13.1 | 385 | 24.3 | 181 | 58.3 | 232 | 14.3 | 224 | 12.4 |
| URTI | 2,212 | 10.9 | 1451 | 9.7 | 292 | 18.4 | 108 | 34.8 | 198 | 12.2 | 163 | 9.0 |
| Croup | 993 | 4.9 | 677 | 4.5 | 100 | 6.3 | 49 | 15.8 | 63 | 3.9 | 104 | 5.7 |
| LRTI | 493 | 2.4 | 288 | 1.9 | 71 | 4.5 | 53 | 17.1 | 40 | 2.5 | 41 | 2.3 |
| Influenza | 478 | 2.4 | 292 | 2.0 | 60 | 3.8 | 39 | 12.6 | 42 | 2.6 | 45 | 2.5 |
| Bronchitis | 296 | 1.5 | 181 | 1.2 | 34 | 2.1 | 10 | 3.2 | 25 | 1.5 | 46 | 2.5 |

Reference group = appropriate-for-gestational age infants born at term; LGA = large for gestational age; LRTI = lower respiratory tract infections; URTI = upper respiratory tract infections; PTB = preterm birth; SGA small for gestational age.

compared to the reference group. Hospitalization rates for bronchiolitis (aRR 1.13; 95% CI 1.01, 1.27) and other URTI (aRR 1.25; 95% CI 1.05, 1.49) were significantly higher among children born SGA relative to the reference group.

Children born LGA at term, had increased ED visit rates for croup (aRR 1.17; CI 1.11, 1.24) and other URTI (aRR 1.03; 95% CI 1.01–1.04), and higher hospitalization rates for bronchiolitis (aRR 1.14; 95% CI 1.02, 1.26) and bronchitis (aRR 1.86; 95% CI 1.35, 2.57) relative to the reference group.

### Age-specific trajectories of total respiratory health service utilization rates

Age-specific trajectories of total respiratory health service utilization rates from birth to five years of age among the different adverse birth groups are presented in Table 3 and Fig 4. All trajectories followed a decreasing trend in the rates of respiratory health service utilization from birth to five years of age.

Compared to the reference group, children born very PTB, moderate/late PTB, and LGA at term significantly required more respiratory health services over time. Children born SGA at term had similar respiratory health services utilization rates as the reference group at birth and at five years of age, and small reductions of health services utilization rates at ages 1 to 2 (-6%), 2 to 3 (-8%) and 3 to 5 (-7% each year).

### Discussion

Using a population-based retrospective cohort design, this study found patterns of respiratory health services utilization in the first five years of life that were distinctive for the group of adverse birth outcomes evaluated. While PTB accounted for the highest rates of ED visits and/

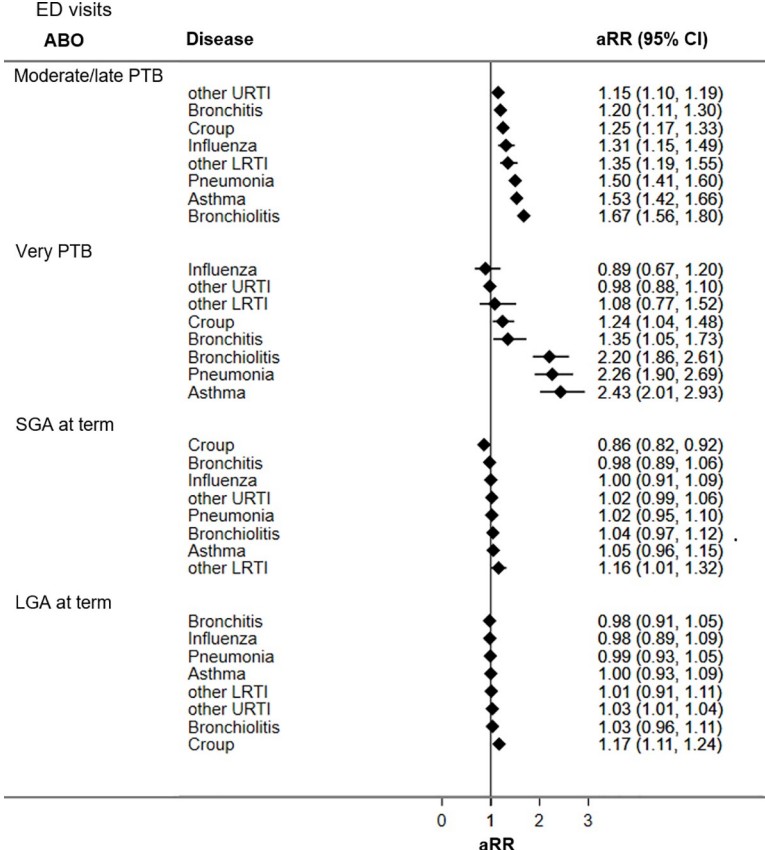

**Fig 2. Adjusted rate ratios of respiratory hospitalizations for adverse birth groups.** CI = confidence interval; LGA = large for gestational age; LRTI = lower respiratory tract infections; URTI = upper respiratory tract infections; PTB = preterm birth; SGA = small for gestational age. Reference group = appropriate-for-gestational age infants born at term; aRR = rate ratio adjusted for sex, bronchopulmonary dysplasia, Apgar 5, score, use of significant resuscitative measures, and material and social deprivation.

or hospitalizations for all respiratory conditions evaluated, SGA at term associated with higher rates of ED visits for other LRTI, higher hospitalizations rates for bronchiolitis and other URTI, but lower ED visit rates for croup. LGA at term resulted in higher rates of ED visits for croup and other URTI and for bronchitis and bronchiolitis hospitalizations. Age-specific trajectories of respiratory health services utilization rates throughout the first five years of life also showed distinct patterns among the adverse birth groups: increased rates among children born PTB, moderately high for children LGA at term, and similar or lower rates for SGA compared to the reference group.

The observed decline in respiratory health service utilization after the first two years of life for all analyzed groups is consistent with prior literature as lung development increases alveolarization during the first 2–4 years of life [41]. Our results showed that children who experienced PTB (moderate and very-PTB) and LGA had higher rates of respiratory health services utilization for the first 5 years compared to the reference group, suggesting that there is room for more preventive management to help families keep these children healthy at home. Targeted public health interventions are required to promote health and minimize respiratory illnesses in these babies to improve morbidity and decrease health care costs. As well, knowing that a patient is in a higher risk group such as LGA will help to direct strategies for parent/caregiver counseling.

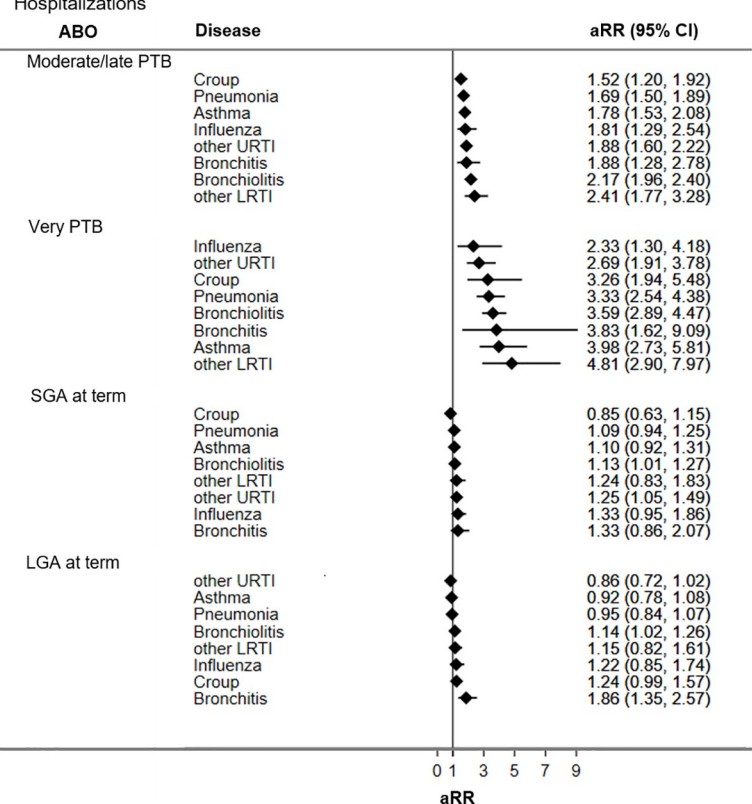

**Fig 3. Adjusted rate ratios of respiratory ED visits for adverse birth groups.** CI = confidence interval; LGA = large for gestational age; LRTI = lower respiratory tract infections; URTI = upper respiratory tract infections; PTB = preterm birth; SGA = small for gestational age. Reference group = appropriate-for-gestational age infants born at term; aRR = rate ratio adjusted for sex, bronchopulmonary dysplasia, Apgar 5, score, use of significant resuscitative measures, material and social deprivation.

Our study results about the large impact of PTB on lung health in early childhood are consistent with a growing body of evidence summarized in meta-analyses of observational studies about the strong association between PTB and asthma [8,42–44]. Children born PTB in our study had higher hospitalization rates for pneumonia and bronchiolitis, conditions that are frequently caused by respiratory viral infections. Importantly, our study adds evidence of the strong association between PTB and respiratory infections in early childhood. Compared to the reference group, very PTB children had a 4-fold increase in the hospitalization rate for other LRTI and a 3-fold increase in the rate of hospital admissions for croup. These results are consistent with other reports of increased risk of respiratory infections among PTB children [14] due to impaired lung development and defects in the immune system [45] that are accentuated by decreasing gestational age [46].

The high rates of other LRTI and bronchiolitis ED visits and other URTI hospitalizations among children born SGA at term align with other population-based studies showing a high risk of respiratory hospitalizations in SGA children (including those born prematurely) during early childhood [14]. Our results also align with those reported by Yoshimoto et al. [15], who did not find a significant association between SGA at term and an increased risk of bronchitis, pneumonia, and asthma hospitalizations at five years of age. However, we found high rates of hospitalizations for bronchiolitis, suggesting that SGA at term could be associated with inflammation of the small airways. Our results and those of earlier studies suggest a vulnerability in

**Table 3. Adjusted rates and rate differences for age-specific trajectories of respiratory health service utilization for birth groups.**

| Age (in years) | Birth group | Adjusted Rate [95% CI] | Rate difference [95% CI] | Percentage of change (% [95% CI]) |
|---|---|---|---|---|
| Birth to < 1 | Reference | 370.7 [367.6, 373.8] | Reference | |
| | Moderate/late PTB | 536.9 [524.1, 549.8] | 166.2 [147.3, 185.2]* | 45 [41, 47] |
| | Very PTB | 569.1 [537.1, 601.0] | 198.4 [152.4, 244.3]* | 54 [45, 61] |
| | SGA at term | 357.0 [347.9, 366.2] | -13.7 [-27.5, 0.1] | -4 [−6, 0] |
| | LGA at term | 433.1 [423.4, 442.7] | 62.4 [47.9, 76.9]* | 17 [14, 18] |
| 1 to < 2 | Reference | 345.3 [342.3, 348.3] | Reference | |
| | Moderate/late PTB | 454.0 [442.2, 465.8] | 108.7 [91.3, 126.2]* | 31 [29, 34] |
| | Very PTB | 590.6 [558.0, 623.1] | 245.3 [198.4, 292.2]* | 71 [63, 79] |
| | SGA at term | 325.8 [317.1, 334.6] | -19.5 [-32.7, -6.2]* | - 6 [−7, −4] |
| | LGA at term | 395.2 [386.0, 404.4] | 49.9 [36.0, 63.8]* | 14 [13, 16] |
| 2 to < 3 | Reference | 228.5 [226.0, 230.9] | Reference | |
| | Moderate/late PTB | 288.0 [278.6, 297.4] | 59.5 [45.7, 73.4]* | 26 [23, 29] |
| | Very PTB | 380.8 [355.0, 406.7] | 152.3 [115.2, 189.6]* | 67 [57, 76] |
| | SGA at term | 211.0 [204.0, 218.1] | -17.5 [-28.1, -6.8]* | - 8 [−10, −6] |
| | LGA at term | 251.9 [244.6, 259.3] | 23.4 [12.3, 34.5]* | 10 [8, 12] |
| 3 to < 4 | Reference | 181.9 [179.8, 184.1] | Reference | |
| | Moderate/late PTB | 235.4 [226.9, 243.9] | 53.5 [40.9, 66.0]* | 29 [26, 33] |
| | Very PTB | 247.9 [227.2, 268.6] | 66 [36.1, 95.8]* | 36 [26, 46] |
| | SGA at term | 168.9 [162.6, 175.2] | -13 [-22.6, -3.5]* | - 7 [−9, −5] |
| | LGA at term | 206.5 [199.9, 213.2] | 24.6 [14.6, 34.6]* | 14 [11, 16] |
| 4 to < 5 | Reference | 147.9 [146.0, 149.9] | Reference | |
| | Moderate/late PTB | 186.6 [179.0, 194.2] | 38.7 [27.4, 49.8]* | 26 [23, 29] |
| | Very PTB | 230.3 [210.4, 250.3] | 82.4 [53.7, 111.1]* | 56 [44, 67] |
| | SGA at term | 137.4 [131.7, 143.1] | -10.5 [-19.2, -2.0]* | - 7 [−10, −5] |
| | LGA at term | 158.2 [152.4, 164.1] | 10.3 [1.5, 19.1]* | 7 [4, 9] |
| 5 to < 6 | Reference | 115.4 [113.7, 117.1] | Reference | |
| | Moderate/late PTB | 143.3 [136.7, 149.9] | 27.9 [18.1, 37.7]* | 24 [20, 28] |
| | Very PTB | 157.5 [141.1, 173.9] | 42.1 [18.4, 65.8]* | 36 [24, 49] |
| | SGA at term | 111.1 [106.0, 116.2] | -4.3 [-12.0, 3.4] | -4 [−7, 3] |
| | LGA at term | 133.4 [128.0, 138.7] | 18 [9.9, 26.1]* | 16 [12, 19] |

Reference group = appropriate-for-gestational age infants born at term; CI = confidence interval; LGA = large for gestational age; PTB = preterm birth; SGA = small for gestational age.

Adjusted rates expressed as aggregated ED visits and hospitalizations for all combined respiratory conditions per 1,000 singleton live births.

* statistically significant differences (p-value < 0.05) between the corresponding adverse birth group and the reference group using Bonferroni's correction of p-value for multiple comparisons.

respiratory function likely associated with impaired immune responses to infection in children with low weight at birth [45], either because of nutritional deprivation [47] or lung abnormalities that predispose to airway inflammation [48]. We did not observe increased rates for asthma among children born SGA at term compared to the reference group. It has been previously reported that SGA preterm children have an increased risk of developing asthma when low birth weight (< 2500 g) is used as an indicator of intrauterine growth restriction [49]. Our results align with those of a population-based study that did not find associations between SGA and asthma in children aged less than 10 years of age [50]. Additionally, the similar and even lower age-trajectories rates for the combined respiratory conditions for SGA at term compared to the reference group may be explained by the inclusion of exclusively SGA at term

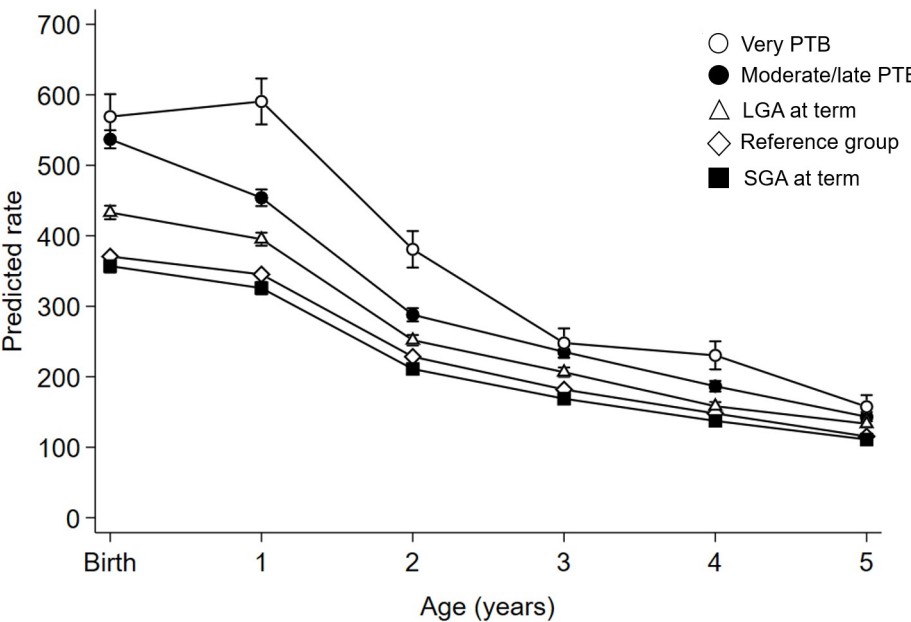

**Fig 4. Age-specific trajectories of respiratory health service utilization rates from birth to five years of age for birth groups.** Reference group = appropriate-for-gestational age infants born at term; LGA = large for gestational age; PTB = preterm birth; SGA = small for gestational age. Rates expressed as number of events per 1,000 singleton live births. Bars express 95% confidence intervals.

in the case definition. Most of the studies reporting SGA include SGA preterm cases, which limits comparability with our results. Another potential explanation for the small differences between rates may be related to the population growth chart that was used to classify SGA cases. The 2001 Canadian population growth chart likely classify small babies of certain immigrant populations as SGA cases. It is known that immigration in Alberta has been growing considerably since 2000 [51], and Heaman et al. [52] reported an odds ratio of 1.7 (95%CI: 1.1–2.4) for SGA related to recent immigrants to Canada.

Previous research suggests that children born LGA are at a high risk of developing respiratory distress at birth [11]. We found higher rates of respiratory health service utilization among LGA children for upper tract infections and bronchiolitis, especially during the first two years of life. Finally, children born LGA at term in our study did not have higher rates of asthma health services utilization. This result is consistent with findings from a recent meta-analysis of 90,000 children and adults reporting no association between birth weight > 4 kg and subsequent risk of asthma [49]. Physiological pathways leading to respiratory problems in LGA babies remain unclear. It has been observed that LGA neonates have an increased risk of respiratory distress at birth [11], but it has also been reported that infants born large for gestational age are more likely to be obese [53]. A high prevalence of childhood obesity (around 30%) has been reported in Canada [54]. Obese children have several complications including breathing difficulties [12,53]. This can be a potential explanation for the respiratory trajectories among LGA cases. Lack of longitudinal weight gain data in our research precluded exploration of the extent of respiratory morbidity among LGA cases that is associated with overweight.

Strengths of this study include the use of a validated perinatal clinical registry to identify the adverse birth exposure groups. The linkage of population-based administrative health data allowed us to assemble a large cohort of children born in Alberta over a 5-year period (2005–

2010) and follow up their history of respiratory health services utilization over the first five years of their life.

Study limitations are related to potential misclassification of respiratory outcomes. We used ICD diagnostic codes recorded in administrative databases to estimate rates of respiratory health services utilization. The impact of potential misclassification in outcome assessment is expected to be minimal because the healthcare databases used in this study follow high-quality data protocols [55]. A set of quality control measures to ensure high-quality hospitalization data are applied by the Canadian Institute for Health Information such as examining relationships between data elements and element edits to each abstract [55]. However, limitations of using single code diagnoses data for some conditions like asthma exist, which is difficult to diagnose using single coding administrative data [56] especially during early childhood. There is a potential overestimation of both hospitalizations and ED visit rates as we were not able to adjust the denominators for the number of children who moved out of the province during the follow-up period due to lack of migration data. However, as a low annual average rate of emigration during the study period has been estimated (around 0.18% [57]), the impact of migration on rate calculations is likely low. Residual confounding is expected to some extent since the datasets do not register clinical information about other important factors related to respiratory health (e.g., ventilation therapies or improvements in subsequent health care for neonates experiencing adverse birth outcomes, and breastfeeding). The use of population-level data and hierarchical models at the DA level accounted for possible differential geographical access to health services and climate conditions that may modulate the frequency of some respiratory diseases, helping to reduce bias in RR estimations.

## Conclusion

This study supports evidence about increased ED visits and hospitalizations for respiratory problems in the first five years of life in children born PTB. Additionally, the study showed that alterations in fetal growth, especially in LGA at term cases, increased the use of respiratory health services utilization during early childhood, particularly for respiratory infections. The population-level patterns of age-specific trajectories in health service utilization have the potential to inform the development of targeted paediatric strategies supportive of healthy lung development in children who experienced adverse birth outcomes. Future research using life-course and environmental approaches are needed to improve our knowledge of the relationships between adverse birth outcomes and lung function in early childhood.

## Supporting information

**S1 Checklist. STROBE statement—checklist of items that should be included in reports of *cohort studies*.**
(DOC)

**S1 Fig. Cumulative percentage (%) of the number of ED-visits per child.**
(TIFF)

**S2 Fig. Cumulative percentage (%) of the number of hospitalizations per child.**
(TIFF)

**S1 Table. Rate ratios for respiratory hospitalizations and ED visits from birth to five years for adverse birth groups.**
(PDF)

**S1 Appendix. Age-specific trajectories of respiratory health services utilization for single respiratory conditions by birth groups.**
(PDF)

## Author Contributions

**Conceptualization:** Jesus Serrano-Lomelin, Anne Hicks, Manoj Kumar, David W. Johnson, Radha Chari, Alvaro Osornio-Vargas, Susan Crawford, Jeffrey Bakal, Maria B. Ospina.

**Data curation:** Jesus Serrano-Lomelin, Susan Crawford, Jeffrey Bakal, Maria B. Ospina.

**Formal analysis:** Jesus Serrano-Lomelin, Maria B. Ospina.

**Funding acquisition:** Maria B. Ospina.

**Investigation:** Maria B. Ospina.

**Methodology:** Jesus Serrano-Lomelin, Anne Hicks, Manoj Kumar, David W. Johnson, Radha Chari, Alvaro Osornio-Vargas, Maria B. Ospina.

**Project administration:** Jesus Serrano-Lomelin, Maria B. Ospina.

**Resources:** Susan Crawford, Jeffrey Bakal, Maria B. Ospina.

**Software:** Jesus Serrano-Lomelin, Susan Crawford, Jeffrey Bakal.

**Supervision:** Maria B. Ospina.

**Writing – original draft:** Jesus Serrano-Lomelin, David W. Johnson, Maria B. Ospina.

**Writing – review & editing:** Jesus Serrano-Lomelin, Anne Hicks, Manoj Kumar, Radha Chari, Alvaro Osornio-Vargas, Susan Crawford, Jeffrey Bakal, Maria B. Ospina.

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
