## [Decision Letter · Decision Letter 0]

30 Oct 2020

PONE-D-20-22831

Patterns of respiratory health services utilization from birth to 5 years of children who experienced adverse birth outcomes

PLOS ONE

Dear Dr. Ospina,

Thank you for submitting your manuscript to PLOS ONE. After careful consideration, we feel that it has merit but does not fully meet PLOS ONE’s publication criteria as it currently stands. Therefore, we invite you to submit a revised version of the manuscript that addresses the points raised during the review process.

Please address all specific points raised by the reviewers, especially reviewer 1 and 2. The compilation of recurrent wheeze with asthma and bronchiolitis puts together different disease entities therefore I suggest a differentiated approach to these entities. I apologize for the delay in finding reviewers for your manuscript what is due to the ongoing COVID-19 pandemic.

We look forward to receiving your revised manuscript.

Kind regards,

Harald Ehrhardt

Academic Editor

PLOS ONE

Journal Requirements:

2. In ethics statement in the manuscript and in the online submission form, please provide additional information about the database used in your retrospective study. Specifically, please ensure that you have discussed whether all data were fully anonymized before you accessed them and/or whether the IRB or ethics committee waived the requirement for informed consent. If patients provided informed written consent to have their data used in research, please include this information.

3.We note that you have indicated that data from this study are available upon request. PLOS only allows data to be available upon request if there are legal or ethical restrictions on sharing data publicly. For information on unacceptable data access restrictions, please see http://journals.plos.org/plosone/s/data-availability#loc-unacceptable-data-access-restrictions.

Reviewers' comments:

Reviewer's Responses to Questions

**Comments to the Author**

1. Is the manuscript technically sound, and do the data support the conclusions?

Reviewer #1: Partly

Reviewer #2: No

Reviewer #3: Yes

2. Has the statistical analysis been performed appropriately and rigorously? 

Reviewer #1: Yes

Reviewer #2: Yes

Reviewer #3: Yes

3. Have the authors made all data underlying the findings in their manuscript fully available?

Reviewer #1: Yes

Reviewer #2: No

Reviewer #3: No

4. Is the manuscript presented in an intelligible fashion and written in standard English?

Reviewer #1: Yes

Reviewer #2: Yes

Reviewer #3: Yes

5. Review Comments to the Author

Reviewer #1: The authors present the results of a retrospective cohort study exploring patterns of respiratory health services utilization in early childhood among children with adverse birth outcomes, defined as being born preterm (PTB), small for gestational age at term (SGA), and large for gestational age at term (LGA). Using administrative health data of all singleton live births in Alberta, Canada between 2005-2010, they estimated adjusted rate ratios and 95% confidence intervals for respiratory ED visits and hospitalizations from birth to 5 years of age for asthma, bronchiolitis, croup, influenza, pneumonia, and other acute upper and lower respiratory tract infections. There were 206,994 children in their cohort, representing 293,764 episodes of respiratory care. Compared to children without adverse birth outcomes, children with PTB had the highest rates of ED visits and hospitalizations, followed by those with LGA. Those with SGA at term had lower rates of ED visits and hospitalizations. Those with SGA at term had the lower rates. Health services utilization rates for respiratory health problems declined with the age of the child from birth to age 5 years.

The authors ask an interesting question about the respiratory health of children born preterm and both small and large for gestational age and their utilization of health services in childhood. There are several questions that need to be addressed before the manuscript is suitable for publication.

1. As the authors note, there is a robust literature on respiratory outcomes among infants born preterm. Why include this group in the study? How does this study add to what is already known?

2. Can the authors include a physiologic rationale/hypothesis for why infants born SGA or LGA might have increased risk for respiratory morbidity in childhood?

3. Small note: ABO commonly refers to blood type groups and, as used here as shorthand for adverse birth outcomes, is a bit confusing for readers.

4. The authors describe a “population-based cohort study” – can they comment on the percentage of births in Alberta represented in this cohort sample?

5. How did the authors conceptualize the covariates they included in the analyses? Adjustment implies confounding, though it is not clear how factors like BPD serve as prior common causes of both exposure and outcome. Clarification is needed, as the conceptualization of the role of the covariates informs the appropriateness of the analytic strategies used. For example, BPD is more likely a mediator rather than confounder. Sex may be conceptualized as an effect modifier. A conceptual model or diagram would help the reader understand more clearly how the authors view the relationships among the variables.

6. The authors use a robust measure for SES – the Pampalon Material and Social Deprivation Index. As there are well-described associations between environmental and social risk and respiratory health in children, especially for those with asthma, why did the authors opt to adjust their analyses for this important driver of health? Can they present the effect estimates of the index on rates of admission/ED visits to allow readers to assess the magnitude of effect in relation to that of ABO?

7. The longitudinal modeling is quite interesting and adds quite a bit to the study. Close to 90,000 individuals had more than 1 ED visit or hospitalization during the study period. It would be helpful to know how many children had 1, 2, 3, etc visits over the 5 years. That is, how much information did each child contribute to the longitudinal analysis?

8. Do the rates of admission/ED visits between ABO groups differ significantly in each year?

9. The authors describe their findings regarding children born SGA in the context of existing literature. One wonders why, in their study, the authors found lower rates of respiratory illness-related health services use compared to the other groups of children. Might they discuss why this may be the case? Is there a biologic mechanism at play Or perhaps might this be a result of the modeling approach?

10. The findings regarding children born LGA are also intriguing. Why might respiratory distress at birth – often associated with maternal diabetes or macrosomia-related birth trauma among LGA infants – be linked to risk for respiratory morbidity during childhood in this group of children?

11. The graphical representation of hospitalization/ED visit rates by year show a striking decline between 1 and 5 years of age for all groups. This is consistent with prior literature, as risk for respiratory morbidity falls with age. The authors do not include this finding in their discussion section. What implications do these data have on clinical care and health policy?

Reviewer #2: This population-based retrospective cohort study compared patterns of respiratory ED visits and hospitalizations during the first five years of life of children born with preterm birth, SGA at term or LGA at term, and demonstrated that SGA and LGA at term increased the use of respiratory health services utilization during early childhood, particularly for respiratory infections. Although the respiratory outcomes based on the ICD diagnostic codes were not necessarily accurate, the large sample size was a strength of this cohort study.

Specific comments

1) The criteria to merge recurrent wheezing (R06.2) with asthma (J45) or bronchiolitis (J20-J21) should be specified.

2) Acute bronchitis (J22) could include the patients diagnosed as (J40.1), and (J20). Therefore, acute bronchiolitis (J21) and acute bronchitis (J20) should be separately analyzed.

3) Age-specific trajectories of total respiratory health service utilization rates were analyzed. Incidences of respiratory diseases, especially infectious diseases, are well-known to change depending on the subjects’ age. Therefore, it is more informative to analyze age-specific trajectories of respiratory health service utilization due to bronchiolitis, croup, or asthma, separately.

4) The first paragraph in the discussion section is a mere repetition of the results, so that it should be concisely summarized.

5) The reason why the discrepancy of the association between SGA at term and an increased risk of bronchitis was observed among different studies should be discussed.

Reviewer #3: The manuscript presents an interesting analysis of routinely collected perinatal and healthcare utilisation data from Canada. The data supports some commonly held beliefs around respiratory health in children who are born early, but also challenges some of the beliefs around the respiratory health of those children born either small or large for gestational age. The data are presented clearly and logically. For the most part and I think it adds to our knowledge in this area. The data is not freely available but this is explained in the manuscript.

I have some small comments or suggestions which do not significantly impact the value of the manuscript. I think it would be useful if the authors could make some comment on the quality of data capture and also in their health system whether episodes are likely to have been missed. There is always an inherent inaccuracy in data coding. There is a short comment on how to code recurrent wheezing in the methods. I'm not sure that a European audience would be content with recurrent wheezing being diagnosed as bronchiolitis, with the exception may be of two or three episodes in the first winter in the first year of life.

The most interesting data actually relates to those children born small for gestational age or large for gestational age. There is some mention in the text about beliefs around respiratory outcomes in these children, but the references are perhaps not as complete as they might be. I'm also not entirely sure why multiple births were excluded and the data is restricted to singleton births only.

There are a few typographical errors. There is an error at the beginning of the sentence in line 211. There is also an inappropriate use of the semicolon in line 214. I'm sure there may be one or two others that I have missed.

The terminology or abbreviation for adverse back birth outcomes is the same as that used for ABO blood groups and in the text 'ABO type' is referred to. I can't think of a good alternative for 'adverse birth outcomes' but I would certainly avoid the phrase 'ABO type'.

In the discussion around respiratory health outcomes in children born small for gestation age, emphasis is made around those few diagnoses where healthcare utilisation is increased. Actually the most surprising thing about the data for those children born small for gestational age is how little their respiratory health appears to be affected. I think this could be discussed a little more. Finally, the Forrest plots presented are visually appealing and helpful for the reader. I do wonder if the diagnoses list for each adverse birth outcomes group and the control group should perhaps be in the same order (although this might be difficult to arrange as Forrest plots software tends to present as the authors have shown).

I think this is a useful paper and certainly merits publication

6. PLOS authors have the option to publish the peer review history of their article (what does this mean?). If published, this will include your full peer review and any attached files.

Reviewer #1: No

Reviewer #2: **Yes: **Yusei Ohshima

Reviewer #3: **Yes: **Dr Gary Doherty

---

## [Author Response · Author response to Decision Letter 0]

7 Jan 2021

Editor’s Comments.

Please address all specific points raised by the reviewers, especially reviewer 1 and

2. The compilation of recurrent wheeze with asthma and bronchiolitis puts together

different disease entities therefore I suggest a differentiated approach to these entities.

I apologize for the delay in finding reviewers

Response

Thank you for the opportunity to reply to the reviewers’ comments and for the very constructive feedback. We introduced modifications throughout the manuscript in light of your helpful comments and suggestions. Please see below responses to all comments and line references to where modifications have been made in the revised manuscript. The modified manuscript includes all changes and other minimal changes required after the manuscript revision. 

The compilation of recurrent wheezing with asthma or bronchiolitis has been explained as suggested by reviewer # 2 (point # 13 below). In the acute setting, physicians struggle to come up with accurate diagnostic codes for conditions with overlapping presentations such as bronchiolitis, wheeze and asthma. To complicate matters, there are differing ways in which a diagnosis of asthma is made, and guidelines have evolved over the years. This means that when using diagnostic codes, it is frequent that the three conditions overlap. We have explained the criteria to merge recurrent wheezing with asthma or bronchiolitis, and added three references. See Lines 147-151: “Recurrent wheezing (R06.2) events were merged with asthma or bronchiolitis based on the most prevalent condition after the first wheezing episode. This merge was made because early life diagnosis of asthma and/or bronchiolitis is particularly difficult in acute settings, and recurrent wheezing has been associated with the development of asthma and respiratory viral infections [32-34].”

Reviewer #1

COMMENT 1. As the authors note, there is a robust literature on respiratory outcomes

among infants born preterm. Why include this group in the study? How does this study

add to what is already known?

RESPONSE: We agree with the reviewer that the effects of duration of gestation on early-life

respiratory outcomes have been widely studied. However, comparisons of outcomes

from infants born preterm with those with alterations in fetal growth (SGA, LGA) are

scarce in the scientific literature. We have provided further clarifications about the comparative purpose of our study under objective 2 (Lines 92-95): “(2) To compare age-specific trajectories of total respiratory health services utilization across adverse birth outcomes. Results from this

study may improve our understanding of the relative importance of alterations in fetal

growth and duration of gestation in early childhood healthcare pathways.”

COMMENT 2. Can the authors include a physiologic rationale/hypothesis for why

infants born SGA or LGA might have increased risk for respiratory morbidity in

childhood?

RESPONSE: We have added in the introduction section a paragraph describing a potential

physiologic rationale linking SGA and LGA to respiratory problems during childhood in

Lines 63-67: “For SGA, insufficient input of oxygen and metabolites linked to fetal growth restrictions could negatively impact the lung development of the fetus [2]. For LGA, both the higher risk of experiencing respiratory distress syndrome [11] and the reduced lung functional

capacity observed in obese infants may alter the risk of suffering respiratory problems

[12].“

COMMENT 3. Small note: ABO commonly refers to blood type groups and, as used here as shorthand for adverse birth outcomes, is a bit confusing for readers. 

RESPONSE: We have changed the term adverse birth outcomes instead of the acronym ABO through the manuscript, including figures and tables.

COMMENT 4. The authors describe a “population-based cohort study” – can they comment on the percentage of births in Alberta represented in this cohort sample?

RESPONSE: Our cohort included singleton livebirths, which represented a 96.6% of the total births; the remaining 3.4% of births were multiple births. We have added this information into the Results section (Lines 232-233): “The birth cohort consisted of 206,994 live singleton births (Fig 1), which represent 96.6% of the total births registered in the province during the study period.”

COMMENT 5. How did the authors conceptualize the covariates they included in the analyses? Adjustment implies confounding, though it is not clear how factors like BPD serve as prior common causes of both exposure and outcome. Clarification is needed, as the conceptualization of the role of the covariates informs the appropriateness of the analytic strategies used. For example, BPD is more likely a mediator rather than confounder. Sex may be conceptualized as an effect modifier. A conceptual model or diagram would help the reader understand more clearly how the authors view the relationships among the variables.

RESPONSE: We used the covariates as control variables at baseline. They are treated as explanatory known risk factors associated with respiratory problems during the infant and early childhood stages, regardless of their role in causal pathways. Explaining causal pathways was not part of our research objectives. We aimed to generate measures of association balanced for other known risk factors. We have explained this in the methods’ section (Lines 174-178): “The previous clinical and sociodemographic characteristics were treated as risk factors associated with respiratory problems during early childhood stages regardless of their role in causal pathways (i.e., confounders, moderators, or effect-modifiers). Our comparative analysis was not aimed to formally test causal pathways, but in generating measures of association balanced for other known risk factors.”

COMMENT 6. The authors use a robust measure for SES – the Pampalon Material and Social Deprivation Index. As there are well-described associations between environmental and social risk and respiratory health in children, especially for those with asthma, why did the authors opt to adjust their analyses for this important driver of health? Can they present the effect estimates of the index on rates of admission/ED visits to allow readers to assess the magnitude of effect in relation to that of ABO?

RESPONSE: We have previously estimated the association between the Material and Social Deprivation Index on hospitalizations and ED visits for every single respiratory outcome using the same cohort. Those results were published at “Belon AP, et al. Health gradients in emergency visits and hospitalisations for paediatric respiratory diseases: A population-based retrospective cohort study. Paediatr Perinat Epidemiol. 2020. https://doi.org/10.1111/ppe.12639”. We have added a comment in the methods’ section (Lines 178-180): “Moreover, the essential role of socioeconomic in producing respiratory health differences for each respiratory condition included in this study has been previously evaluated [39].”

COMMENT 7. The longitudinal modeling is quite interesting and adds quite a bit to the study. Close to 90,000 individuals had more than 1 ED visit or hospitalization during the study period. It would be helpful to know how many children had 1, 2, 3, etc. visits over the 5 years. That is, how much information did each child contribute to the longitudinal analysis?

RESPONSE: We have referenced this information in the manuscript body (Lines 252-254): 

“Fifty-eight percent of children had two or more ED visits, while 33% had two or more hospitalizations during the study period. The number of events per child until they turned five years of age are shown in the supplementary S1 and S2 Figs.”

COMMENT 8. Do the rates of admission/ED visits between ABO groups differ significantly in each year?

RESPONSE: We have indicated in the methods section (Lines 213-215): “Finally, for each year of life, we estimated the difference in marginal rates between the adverse birth group and the reference group using contrast tests with Bonferroni’s correction to adjust the 95% CIs”. The results are presented in Table 3. In order to clarify the statistical significance of effects and facilitate comparability of results, we have inserted an asterisk on Table 3 (“*”) to indicate statistically significant differences by year (p-value<0.05 after Bonferroni’s correction). See Lines 330-332: “*statistically significant differences (p-value < 0.05) between the corresponding adverse birth group and the reference group using Bonferroni’s correction of p-value for multiple comparisons.” 

COMMENT 9. The authors describe their findings regarding children born SGA in the context of existing literature. One wonders why, in their study, the authors found lower rates of respiratory illness-related health services use compared to the other groups of children. Might they discuss why this may be the case? Is there a biologic mechanism at play or perhaps might this be a result of the modeling approach?

RESPONSE: We evaluated the rates for SGA at term to avoid the overlap between SGA cases with gestational age before 38 weeks and prematurity. It is probable that, for this reason, we did not find higher rates in the SGA group compared to the reference group. Furthermore, it is probable that the 2001-Canadian population chart used for the SGA case definition classifies small babies from immigrants’ populations (which have been growing in Alberta during since 2000) as SGA cases. We have expanded the Discussion section to address this issue (Lines 391-400)” “Additionally, the similar and even lower age-trajectories rates for the combined respiratory conditions for SGA at term compared to the reference group may be explained by the inclusion of exclusively SGA at term in the case definition. Most of the studies reporting SGA include SGA preterm cases, which limits comparability with our results. Another potential explanation for the small differences between rates may be related to the population growth chart that was used to classify SGA cases. The 2001 Canadian population growth chart likely classify small babies of certain immigrant populations as SGA cases. It is known that immigration in Alberta has been growing considerably since 2000 [51], and Heaman et al. [52] reported an odds ratio of 1.7 (95%CI: 1.1-2.4) for SGA related to recent immigrants to Canada.”

COMMENT 10: The findings regarding children born LGA are also intriguing. Why might respiratory distress at birth – often associated with maternal diabetes or macrosomia-related birth trauma among LGA infants – be linked to risk for respiratory morbidity during childhood in this group of children?

RESPONSE: We have added the following clarification in the Discussion to address the reviewer’s question. Lines 408-415: “Physiological pathways leading to respiratory problems in LGA babies remain unclear. It has been observed that LGA neonates have an increased risk of respiratory distress at birth [11], but it has also been reported that infants born large for gestational age are more likely to be obese [53]. A high prevalence of childhood obesity (around 30%) has been reported in Canada [54]. Obese children have several complications including breathing difficulties [53, 12]. This can be a potential explanation for the respiratory trajectories among LGA cases. Lack of longitudinal weight gain data in our research precluded exploration of the extent of respiratory morbidity among LGA cases that is associated with overweight.”

COMMENT 11: The graphical representation of hospitalization/ED visit rates by year show a striking decline between 1 and 5 years of age for all groups. This is consistent with prior literature, as risk for respiratory morbidity falls with age. The authors do not include this finding in their discussion section. What implications do these data have on clinical care and health policy?

RESPONSE: We have included in the discussion section the potential clinical care implications of this result. Lines 353-362: “The observed decline in respiratory health service utilization after the first two years of life for all analyzed groups is consistent with prior literature as lung development increases alveolarization during the first 2–4 years of life [41]. Our results showed that children who experienced PTB (moderate and very-PTB) and LGA had higher rates of respiratory health services utilization for the first 5 years compared to the reference group, suggesting that there is room for more preventive management to help families keep these children healthy at home. Targeted public health interventions are required to promote health and minimize respiratory illnesses in these babies to improve morbidity and decrease health care costs. As well, knowing that a patient is in a higher risk group such as LGA will help to direct strategies for parent/caregiver counseling.”

Reviewer #2

COMMENT 12. This population-based retrospective cohort study compared patterns of respiratory ED visits and hospitalizations during the first five years of life of children born with preterm birth, SGA at term or LGA at term, and demonstrated that SGA and LGA at term increased the use of respiratory health services utilization during early childhood, particularly for respiratory infections. Although the respiratory outcomes based on the ICD diagnostic codes were not necessarily accurate, the large sample size was a strength of this cohort study.

RESPONSE: Thank you for this comment.

COMMENT 13. The criteria to merge recurrent wheezing (R06.2) with asthma (J45) or bronchiolitis (J20-J21) should be specified).

RESPONSE: In the acute setting, physicians struggle to come up with accurate diagnostic codes for conditions with overlapping presentations such as bronchiolitis, wheeze and asthma. To complicate matters, there are differing ways in which a diagnosis of asthma is made, and guidelines have evolved over the years. This means that when using diagnostic codes, it is frequent that the three conditions overlap. We have explained the criteria to merge recurrent wheezing with asthma or bronchiolitis, and added three references. See Lines 147-151: “Recurrent wheezing (R06.2) events were merged with asthma or bronchiolitis based on the most prevalent condition after the first wheezing episode. This merge was made because early life diagnosis of asthma and/or bronchiolitis is particularly difficult in acute settings, and recurrent wheezing has been associated with the development of asthma and respiratory viral infections [32-34].” 

COMMENT 14. Acute bronchitis (J22) could include the patients diagnosed as (J40.1), and (J20). Therefore, acute bronchiolitis (J21) and acute bronchitis (J20) should be separately analyzed.

RESPONSE: We have re-analyzed both acute bronchitis and acute bronchiolitis separately. We have made several modifications across the manuscript, in the methods section (Line 145), reporting of results (Lines 248-308), including a new Table 2 and forest plots (Figures 2 and 3) and also into the discussion section (Lines 379-383). 

COMMENT 15. Age-specific trajectories of total respiratory health service utilization rates were analyzed. Incidences of respiratory diseases, especially infectious diseases, are well-known to change depending on the subjects’ age. Therefore, it is more informative to analyze age-specific trajectories of respiratory health service utilization due to bronchiolitis, croup, or asthma, separately.

RESPONSE: We originally focused on comparing population-level patterns of age-specific trajectories of total respiratory health services utilization across adverse birth outcomes to help understand the overall burden imposed by adverse birth conditions to respiratory health service use (see lines: 92-93). Based on your comment, we have included supplementary tables to present results of respiratory health service utilization trajectories for single respiratory outcomes (Tables A to H in S1 Appendix). We have also added a new paragraph (Lines 218-220) to report these results: “Age-specific trajectories of respiratory health services utilization for single respiratory conditions by birth groups were estimated and reported in the supplementary material (Tables A to H in S1 Appendix).”

COMMENT 16. The first paragraph in the discussion section is a mere repetition of the results, so that it should be concisely summarized. 

RESPONSE: We have reworded the first paragraph in the discussion to concisely summarize the study results (Lines 341-351): “Using a population-based retrospective cohort design, this study found patterns of respiratory health services utilization in the first five years of life that were distinctive for the group of adverse birth outcomes evaluated. While PTB accounted for the highest rates of ED visits and/or hospitalizations for all respiratory conditions evaluated, SGA at term associated with higher rates of ED visits for other LRTI, higher hospitalizations rates for bronchiolitis and other URTI, but lower ED visit rates for croup. LGA at term resulted in higher rates of ED visits for croup and other URTI and for bronchitis and bronchiolitis hospitalizations. Age-specific trajectories of respiratory health services utilization rates throughout the first five years of life also showed distinct patterns among the adverse birth groups: increased rates among children born PTB, moderately high for children LGA at term, and similar or lower rates for SGA compared to the reference group.” 

COMMENT 17. The reason why the discrepancy of the association between SGA at term and an increased risk of bronchitis was observed among different studies should be discussed.

RESPONSE: We have re-analyzed both acute bronchitis and acute bronchiolitis separately, following the reviewer recommendation in Comment #14. We have reworded a paragraph in the discussion section according to the new results (Lines 379-383): “Our results also align with those reported by Yoshimoto et al. [15], who did not find a significant association between SGA at term and an increased risk of bronchitis, pneumonia, and asthma hospitalizations at five years of age. However, we found high rates of hospitalizations for bronchiolitis, suggesting that SGA at term could be associated with inflammation of the small airways.”

Reviewer #3

COMMENT 18: The manuscript presents an interesting analysis of routinely collected perinatal and healthcare utilisation data from Canada. The data supports some commonly held beliefs around respiratory health in children who are born early, but also challenges some of the beliefs around the respiratory health of those children born either small or large for gestational age. The data are presented clearly and logically. For the most part and I think it adds to our knowledge in this area. The data is not freely available, but this is explained in the manuscript.

RESPONSE: Thank you for this positive feedback.

COMMENT 19. I have some small comments or suggestions which do not significantly impact the value of the manuscript. I think it would be useful if the authors could make some comment on the quality of data capture and also in their health system whether episodes are likely to have been missed. There is always an inherent inaccuracy in data coding. There is a short comment on how to code recurrent wheezing in the methods. I'm not sure that a European audience would be content with recurrent wheezing being diagnosed as bronchiolitis, with the exception may be of two or three episodes in the first winter in the first year of life.

RESPONSE: A reference related to the quality of administrative health data in Canada was originally included in the study limitations’ paragraph (see Lines 425-427), reference#55. 

Based on your comment, we have added a little more about the rationale for high quality data in the Canadian administrative databases, and potential limitations (Lines 427-431): “A set of quality control measures to ensure high-quality hospitalization data are applied by the Canadian Institute for Health Information such as examining relationships between data elements and element edits to each abstract [55]. However, limitations of using single code diagnoses data for some conditions like asthma exist, which is difficult to diagnose using single coding administrative data [56] especially during early childhood.”

In relation to merging wheezing with bronchiolitis or asthma, we have added a rationale in Lines 147-151: “Recurrent wheezing (R06.2) events were merged with asthma or bronchiolitis based on the most prevalent condition after the first wheezing episode. This merge was made because the diagnosis of asthma and/or bronchiolitis early in life is particularly difficult in acute settings, and recurrent wheezing has been associated with the development of asthma and respiratory viral infections [32-34].”

COMMENT 20: The most interesting data actually relates to those children born small for gestational age or large for gestational age. There is some mention in the text about beliefs around respiratory outcomes in these children, but the references are perhaps not as complete as they might be. I'm also not entirely sure why multiple births were excluded, and the data is restricted to singleton births only.

RESPONSE: We have added explanation in Lines 109-111: “Multiple births, representing a small proportion of the total births (approximately 2.6%), were excluded as their neonatal and childhood outcomes are known to differ significantly as compared to the larger proportion of the singleton live births.”

COMMENT 21. There are a few typographical errors. There is an error at the beginning of the sentence in line 211. There is also an inappropriate use of the semicolon in line 214. I'm sure there may be one or two others that I have missed.

RESPONSE: Thank you for this feedback. We have reviewed the manuscript and corrected typographical errors throughout.

COMMENT 22. The terminology or abbreviation for adverse back birth outcomes is the same as that used for ABO blood groups and in the text 'ABO type' is referred to. I can't think of a good alternative for 'adverse birth outcomes' but I would certainly avoid the phrase 'ABO type'.

RESPONSE: We have used adverse birth outcomes groups instead of the ABO acronym. We have made the corresponding changes through the manuscript.

COMMENT 23. In the discussion around respiratory health outcomes in children born small for gestation age, emphasis is made around those few diagnoses where healthcare utilisation is increased. Actually, the most surprising thing about the data for those children born small for gestational age is how little their respiratory health appears to be affected. I think this could be discussed a little more. Finally, the Forrest plots presented are visually appealing and helpful for the reader. I do wonder if the diagnoses list for each adverse birth outcomes group and the control group should perhaps be in the same order (although this might be difficult to arrange as Forrest plots software tends to present as the authors have shown).

RESPONSE: We have sorted the presentation of results in the Forest Plot figures based on the OR to easily observe the respiratory outcomes more affected by each adverse birth condition. As per reviewer recommendation #14, we have re-done the forest plots (Figures 2 and 3) to present separate results for bronchitis and bronchiolitis.

COMMENT 24. I think this is a useful paper and certainly merits publication.

RESPONSE: Thank you for this kind feedback.

---

## [Decision Letter · Decision Letter 1]

9 Feb 2021

Patterns of respiratory health services utilization from birth to 5 years of children who experienced adverse birth outcomes

PONE-D-20-22831R1

Dear Dr. Ospina,

We’re pleased to inform you that your manuscript has been judged scientifically suitable for publication and will be formally accepted for publication once it meets all outstanding technical requirements.

Kind regards,

Harald Ehrhardt

Academic Editor

PLOS ONE

Additional Editor Comments (optional):

Reviewers' comments:

Reviewer's Responses to Questions

**Comments to the Author**

1. If the authors have adequately addressed your comments raised in a previous round of review and you feel that this manuscript is now acceptable for publication, you may indicate that here to bypass the “Comments to the Author” section, enter your conflict of interest statement in the “Confidential to Editor” section, and submit your "Accept" recommendation.

Reviewer #1: All comments have been addressed

Reviewer #2: All comments have been addressed

Reviewer #3: All comments have been addressed

2. Is the manuscript technically sound, and do the data support the conclusions?

Reviewer #1: Yes

Reviewer #2: Yes

Reviewer #3: Yes

3. Has the statistical analysis been performed appropriately and rigorously? 

Reviewer #1: Yes

Reviewer #2: Yes

Reviewer #3: Yes

4. Have the authors made all data underlying the findings in their manuscript fully available?

Reviewer #1: No

Reviewer #2: Yes

Reviewer #3: Yes

5. Is the manuscript presented in an intelligible fashion and written in standard English?

Reviewer #1: Yes

Reviewer #2: Yes

Reviewer #3: Yes

6. Review Comments to the Author

Reviewer #1: The authors have addressed all questions satisfactorily and completely. The manuscript is now suitable for publication.

Reviewer #2: (No Response)

Reviewer #3: I feel the authors have mostly addressed my minor comments, and also appear to have taken on board the comments of the other reviewers.

7. PLOS authors have the option to publish the peer review history of their article (what does this mean?). If published, this will include your full peer review and any attached files.

Reviewer #1: No

Reviewer #2: **Yes: **Yusei Ohshima

Reviewer #3: **Yes: **Dr Gary M. Doherty

---

## [Editor Report · Acceptance letter]

11 Feb 2021

PONE-D-20-22831R1 

Patterns of respiratory health services utilization from birth to 5 years of children who experienced adverse birth outcomes 

Dear Dr. Ospina:

I'm pleased to inform you that your manuscript has been deemed suitable for publication in PLOS ONE. Congratulations! Your manuscript is now with our production department. 

Kind regards, 

on behalf of

Prof. Harald Ehrhardt 

Academic Editor

PLOS ONE